# Left Atrial Functional Remodeling in Patients with Chronic Heart Failure Treated with Sacubitril/Valsartan

**DOI:** 10.3390/jcm12031086

**Published:** 2023-01-30

**Authors:** Michele Correale, Michele Magnesa, Pietro Mazzeo, Martino Fortunato, Lucia Tricarico, Alessandra Leopizzi, Adriana Mallardi, Raffaele Mennella, Francesca Croella, Massimo Iacoviello, Matteo Di Biase, Natale Daniele Brunetti

**Affiliations:** 1Cardiology Unit, Cardio-Thoracic Department, Policlinico Riuniti University Hospital, Viale Luigi Pinto 1, 71100 Foggia, Italy; 2Department of Medical and Surgical Sciences, University of Foggia, 71100 Foggia, Italy

**Keywords:** left atrial function, chronic heart failure reduced ejection fraction, cardiac reverse remodeling, left atrial strain, ARNI, sacubitril/valsartan

## Abstract

(1) Background: Previous studies showed left ventricular (LV) and left atrial (LA) improvement and reverse remodeling after therapy with Sacubitril/Valsartan (S/V) in patients affected by heart failure with reduced ejection fraction (HFrEF). Therefore, we sought to investigate predictors of LA structural and functional reverse remodeling (LARR) in this setting of patients after therapy with S/V, focusing on left atrial strain parameters, such as peak atrial longitudinal strain (PALS). (2) Methods: Patients with HFrEF underwent clinical and echocardiographic evaluation at baseline and after six months of therapy with S/V. Measures of LA structure (LA volume index, LAVi) and function (LA emptying fraction (LAEF), PALS, LA conduit strain and peak atrial contraction strain (PACS) were also analyzed. Patients were divided in two groups, those with a LARR (relative reduction in LAVi > 15%, LARR+) and those without (LARR−). (3) Results: A total of 47 consecutive patients (66 ± 8 years, 85% male, mean LVEF 28 ± 6%) were enrolled in the study and followed up. A significant increase of LAEF (46 ± 13 vs. 37 ± 11%, *p* < 0.001) and a significant reduction of LAVi (42 ± 15 vs. 45 ± 15 mL/m^2^, *p* = 0.008) were found after 6 months of S/V therapy; 47% of the population showed LA reverse remodeling. LA strain parameters, PALS (19 ± 8 vs. 15 ± 7 %, *p* < 0.001) and LA conduit (−9.7 ± 5.2% vs. −7.6 ± 4.1%, *p* = 0.007) significantly improved after 6 months of S/V therapy. At multivariable stepwise regression analysis, changes in LV End Diastolic Volume (LVEDV) and PALS were significantly proportional to changes in LAVi values. (4) Conclusions: Six months of treatment with S/V in patients with HFrEF was associated with an improvement in LA functional reverse remodeling in a real-world scenario. LARR was not significantly correlated to baseline echocardiographic variables, but was proportional to changes in LV volumes and LA strain parameters. Finally, after S/V therapy, a strict connection between LA and LV reverse remodeling and between LA anatomical and functional reverse remodeling seems to be outlined.

## 1. Introduction

Heart failure (HF) is the one of the leading causes of cardiovascular death worldwide; HF progression is linked to cardiac remodeling, a process characterized by myocardial fibrosis [1], decreased cardiac contractility and increased cardiac stiffness, potentially leading to severe arrhythmias. Current therapeutic strategies for HF patients, ACE-inhibitors (ACE-I), angiotensin-receptor blocker (ARBs), beta-blockers and mineralocorticoid receptor antagonists (MRAs) may partly reverse ventricular fibrosis [2,3].

More recently, the use of angiotensin receptor/neprilysin inhibitors (ARNI) was recommended for the treatment of HF with reduced ejection fraction (HFrEF). Results from the PARADIGM-HF trial revealed that treatment with sacubitril/valsartan (S/V) in patients with chronic HF (CHF) and HFrEF was associated with a 20% reduction in cardiovascular deaths or hospitalization compared with enalapril [4].

Recent studies have demonstrated that S/V treatment might improve left ventricular (LV) systolic and diastolic function [5] and reverse remodeling [6] in subjects with HFrEF; S/V may also reduce hospitalization costs in real-world registries [7,8].

Data suggest that ARNI may have antiarrhythmic effects, either by limiting cardiovascular-disease-related proarrhythmic remodeling or through direct antiarrhythmic effects on cardiomyocytes. ARNI modulate cardiac electrophysiology at various scales and affect several determinants of both atrial and ventricular tachyarrhythmias [9]. Several studies reported that S/V does impact the risk of ventricular tachyarrhythmias in chronic HFrEF patients over follow up, showing less sudden cardiac death (SCD), ventricular arrhythmias and appropriate ICD therapy in patients in treatment with S/V [10]. S/V may lead to significant clinical and functional improvements even in HFrEF patients with ICD at greater arrhythmic risk [11]. S/V reduced SCD risk particularly in ICD users and nonischemic cardiomyopathy [12]. Natriuretic peptides (NPs) promote diuresis, natriuresis and vasodilation in early chronic HF, countering renin–angiotensin–aldosterone system (RAAS) and sympathetic nervous system (SNS) overstimulation. Despite important increases in circulating NP levels as CHF progresses, their effects are reduced. In fact, there is reduced availability of active forms of NPs (defined BNP) and the target organ responsiveness is reduced. Finally, the counter-regulatory hormones of the RAAS and SNS are over-activated. The treatment with S/V is associated with increased levels of NPs and their intracellular mediator cyclic guanosine monophosphate, suggesting improved functional effectiveness of the NPs [13]. The combined action of S/V on NPs and RAAS improves the imbalance between the RAAS and NPs [14]. The impact of S/V according to etiology and age was also reported. Particularly, in subanalysis of PARADIGM-HF [15] adjusted outcomes were similar across etiologic categories, as was the benefit of sacubitril/valsartan over enalapril. In a retrospective analysis, side effects were more common in older (>65 years of age) patients [16].

Given the large variability of clinical presentation in patients with equally reduced LVEF [17], the analysis of other cardiac chambers may be useful for a better definition of individual risk and prognosis. Left atrial (LA) dimension and function play a pivotal role in the pathophysiology of HFrEF [18]. LA dysfunction may be the major driver/mediator of clinical decompensation in HF [17]. The recovery of LA function has been defined as LA reverse remodeling (LARR), described by Donal et al. [19] as structural reverse remodeling >15% reduction in LA end-systolic volume index and subsequently reported by Thomas et al. in a state-of-the-art paper [20]. Other parameters, such as LAEF, PALS, LA conduit strain and PACS, are also linked to LA functional reverse remodeling [21].

Therefore, the aim of this study was to analyze the role of PALS and other echo parameters in predicting LARR in patients with HFrEF treated with S/V.

## 2. Methods

Forty-seven consecutive patients with HFrEF in New York Heart Association (NYHA) functional class II-III enrolled in the Daunia Heart Failure Registry as reported elsewhere [22,23,24] were followed up with between September 2019 and March 2020. Therapy with SGLT2i was not yet recommended in the HF guidelines at the time of recruitment. Enrolment criteria included LVEF ≤35%, systolic blood pressure ≥100 mmHg, estimated glomerular filtration rate (eGFR) ≥ 30 mL/min/1.73 m² and potassium levels ≤ 5.4 mEq/L. All patients were treated with stable ACE-inhibitor or angiotensin receptor antagonist doses and started treatment with S/V therapy as recommended by the ESC guidelines on HF diagnosis and treatment [25]. The starting dose was 49/51 mg b.i.d., titrated to 97/101 b.i.d. after at least 2–4 weeks (time of creatinine, potassium, urea and blood pressure assessment) if tolerated. In patients with history of hypotension or with low blood pressure values, the starting dose was 24/26 mg b.i.d. Medical history, heart rate, systolic blood pressure, body mass index, NYHA functional class and medications were recorded and monitored. All patients underwent blood analysis, ECG, conventional and advanced echocardiography with tissue Doppler imaging (TDI) and speckle tracking echocardiography for the assessment of LA function in an ambulatory setting under resting conditions at the beginning and after 6 months of therapy with S/V.

Patients were divided in two groups, those with a LARR (relative reduction in LAVi > 15% [26], LARR+) and those without (LARR−) (Appendix A).

### 2.1. Echocardiography

Conventional echocardiography was used to assess LV dimensions and LVEF, peak velocities of trans-mitral early (E) and late diastolic (A) LV filling, the ratio of trans-mitral early to late (E/A ratio) LV filling velocity. LV dimensions and LVEF were calculated according to recommendations in the joint ASE/ESC guidelines [27]. LVEF was calculated according to the Simpson’s rule.

LA volumes were measured using the biplane method of disks from standard apical 2- and 4-chamber views at end systole (LAVi) and end diastole (LAEDVi). LA borders, traced by planimetry, consisted of the walls of the LA and a line drawn across the mitral annulus. Pulmonary vein ostia and the LA appendage were excluded from the measurement. The moments of first mitral valve opening and closing were used to determine end systole and end diastole, respectively. These measurements were made according to the currently accepted standard for 2D-TTE LA volume measurement [27]. Volumes were indexed using body surface area expressed in m^2^. According to Mosteller’s “simplified calculation of body-surface area in metric terms” the body surface area = the square root of product of the weight in kg times the height in cm divided by 3600.

Pulsed Doppler mitral inflow velocities were obtained by placing a 1–2 mm sample volume between the tips of the mitral leaflets in the apical four-chamber view. The Doppler beam was aligned parallel to the flow direction. TDI measurements were recorded at the septal and lateral mitral annulus in apical four-chamber view, including early (e’) diastolic velocities. The trans-mitral to mitral annular early diastolic velocity ratio (E/e’) was also calculated.

Transthoracic echocardiography was performed using an EPIQ 7C ultrasound system with X5-1 matrix array transducer (Philips Healthcare, Philips Medical Systems, Andover, MA, USA). All echocardiographic studies were performed and interpreted by two experienced physicians who were blinded to the clinical data. Inter-observer concordance in data interpretation was >95%.

### 2.2. Speckle-Tracking Strain Analysis for Assessment Left Atrial Function

Digital Imaging and Communications in Medicine (DICOM) formatted file clips were uploaded into a personal computer for subsequent off-line strain analysis with AutoStrain application (QLAB 10.5 software, Philips Ultrasound, WA, USA). For standardization, according to the recommendations of the EACVI/ASE/Industry Task Force [28] for tracing the LA, a non-foreshortened apical four-chamber view was acquired, starting the tracing at the endocardial border of the mitral annulus and the LA endocardial border was traced, extrapolating across the pulmonary veins and/or LA appendage orifices, up to the opposite mitral annulus side [29].

The LA cardiac cycle consists of three phases: The reservoir phase goes from end-diastole (ED) to end systole (ES) (PALS), the conduit phase (LA conduit strain) ends at the time point right before atrial contraction, which is the third phase that completes the cardiac cycle (LA strain during contraction phase, PACS). As the atrial wall lengthens during the reservoir phase, the strain in this phase should be reported as a positive value (PALS). The shortening of the LA wall during the other two phases suggests that they should be characterized by negative values (LA conduit strain and PACS). Ventricular end-diastole is recommended as the time reference to define the zero-baseline for LA strain curves.

### 2.3. Statistical Analysis

Continuous variables were expressed as mean ± standard deviation (SD) and compared with Student’s *t*-test or Mann–Whitney U test as required, categorical variables as percentages and compared with χ2 test. Mean values were compared with Student’s *t*-test for paired groups or Wilcoxon’s matched pairs test as required. Linear correlations were determined by measuring the Pearson’s correlation coefficient. Multivariable stepwise correction analysis was used to assess possible bias of confounders. A *p* < 0.05 was considered as statistically significant. With a type I error of 0.05 and a type II error (power) of 0.80, a ratio of 2:1 (LARR+ vs. LARR−), with the current population enrolled, a difference between rates of at least 35% could be detected as statistically significant. Data were analyzed with di SPSS 26 statistics software.

## 3. Results

Patient’s characteristics (mean age 66 ± 8, male gender 85%, mean LVEF 28 ± 6%) are given in Table 1; 34% patients were titrated with the maximum dose of S/V 97/103 mg b.i.d., 30% of patients were in therapy with 49/51 mg b.i.d. and the remaining 36% of patients were in therapy with 24/26 mg b.i.d after six months of treatment. These differences in S/V titration doses are influenced by low blood pressure, hyperkalemia and others clinical factors. NYHA class and NT-proBNP levels improved after six months of S/V therapy compared to baseline (2.1 ± 0.4 vs. 2.4 ± 0.5, *p* = 0.006; 399 ± 325 vs. 1086 ± 1116 mg/dL, *p* < 0.001, respectively). Potassium levels were significantly increased after 6 months of therapy, as expected, but no patient experienced hyperkalemia.

### 3.1. Conventional Echo Parameters

A reduction of LVEDVi (99 ± 38 vs. 105 ± 34 mL/m^2^, *p* = 0.07) and LVESVi (65 ± 29 vs. 76 ± 28 mL/m^2^, *p* < 0.001), and an increase of LVEF (36 ± 7% vs. 28 ± 6%, *p* < 0.001) were found at follow up compared to baseline (Table 2). Moreover, a significant change for LAEF (46 ± 13% vs. 37 ± 11%, *p* < 0.001) and for the LAVi (42 ± 15 vs. 45 ± 15 mL/m^2^, *p* = 0.008) (Table 2 and Table 3) were found. E/e’ ratio and grading of LV diastolic dysfunction were also significantly reduced (13 ± 6 vs. 16 ± 6, *p* < 0.001; 1.6 ± 0.7 vs. 1.8 ± 0.9, *p* < 0.001, respectively). The grade of mitral regurgitation was not significantly reduced (1.5 ± 0.6 vs. 1.6 ± 0.7, *p* = 0.07). Finally, 36% of population showed LARR.

### 3.2. Two-Dimensional Speckle Tracking Echocardiography (2D-STE) Parameters

LA strain parameters, PALS (19 ± 8 vs. 15 ± 7 %, *p* < 0.001) and LA conduit (−9.7 ± 5.2 vs. −7.6 ± 4.1%, *p* = 0.007), were found significantly improved after six months of therapy with S/V (Table 3).

### 3.3. Correlations

Changes in LAVi values (6 months vs. baseline) were proportional to:
(i)baseline LVEDD (*r* = 0.3, *p* = 0.03) (Figure 1);(ii)baseline LVESD (*r* = 0.3, *p* = 0.03) (Figure 1);(iii)changes in LVEDV (*r* = 0.48, *p* = 0.001) (Figure 1);(iv)changes in LVESV, (*r* = 0.46, *p* = 0.001);(v)changes in PALS values (*r* = 0.32, *p* = 0.03) (Figure 1);(vi)changes in right ventricle tissue Doppler imaging s’ wave (RV TDI s’; *r* = −0.32, *p* = 0.04) (Figure 1);

Changes in PALS values were proportional to changes in:
(i)LVEF (*r* = 0.45, *p* = 0.002) (Figure 1);(ii)TAPSE (*r* = −0.38, *p* = 0.01) (Figure 1).

**Figure 1 jcm-12-01086-f001:**
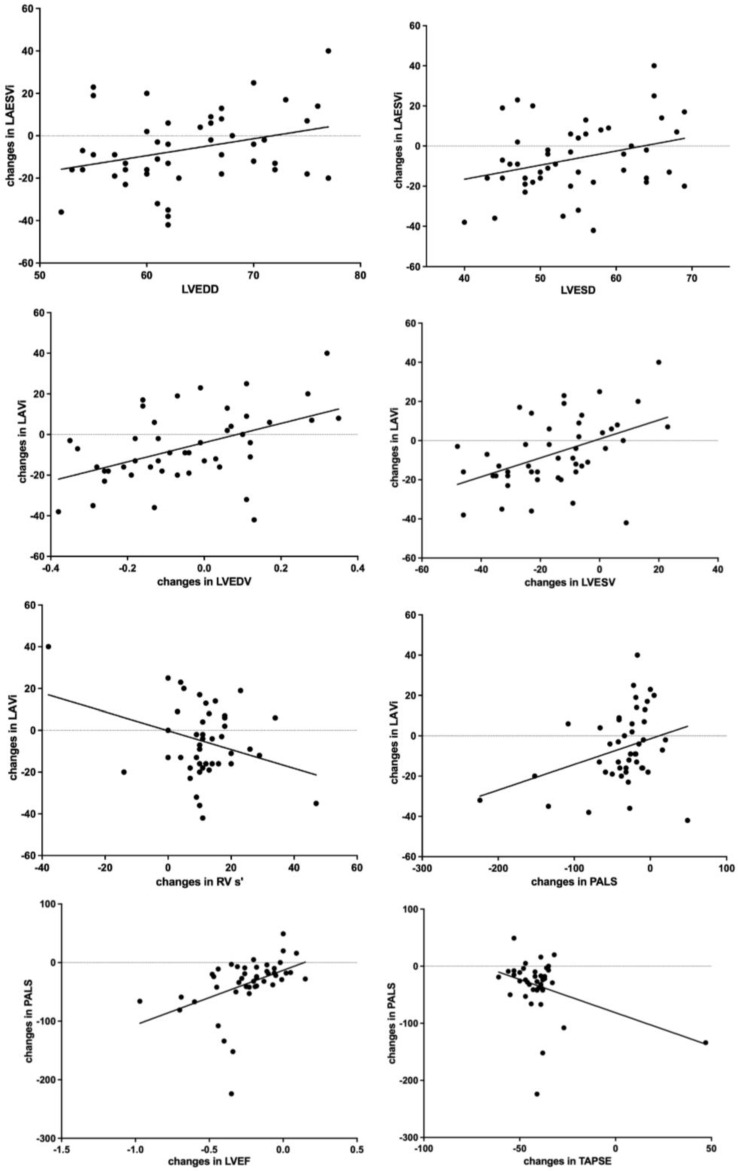
Correlations between changes in LEASVI and LVEDD, LVESD, changes in LVEDV, changes in RV S’ and changes in PALS, and between changes in PALS and changes in LVEF and TAPSE. (Changes are expressed as %, LVEDD and LVESD in mm). I del.

At multivariable stepwise regression analysis, in a model including age, gender, baseline NT-proBNP levels, drug doses after titration, systolic blood pressure, baseline LVEDD and LVESD values, changes in RV TDI s’, LVEDV, LVESV and PALS values, only changes in LVEDV and PALS remained significantly proportional to changes in LAVi values (Appendix A).

### 3.4. LARR+ vs. LARR− Comparison

Comparing basal echocardiographic parameters between the two groups, we observed a positive trend in RV function parameters (TAPSE and RV TDI s’) in predicting LARR before S/V therapy, with a statistical significance for RV FAC (Table 4). In other words, we suggest that patients who initiate S/V therapy and have better RV function are more likely to develop LARR. Evaluating S/V doses in the two subgroups (LARR+ vs. LARR−), we noted that the maximum dose of 97/103 mg at 6 months follow-up was higher in LARR+ than in LARR− patients (Appendix A, *p* = 0.0368) PALS was significantly improved in both groups compared to baseline (*p* < 0.001; Figure 2), but the LARR+ group showed an improvement of the PALS that was twice as high (PALS changes: 55 ± 66% vs. 25 ± 26% *p* = 0.0316).

## 4. Discussion

To the best of our knowledge, this is one of the few studies showing:Significant improvement of LA dimensions and function already after just six months of therapy with S/V;Significant correlations between changes in LA volumes (LAVi) and LV dimensions (basal LVEDD, basal LVESD; changes in LVEDV, changes in LVESV), LA function (PALS) and RV function (changes in RV TDI S’);Changes in LA function (PALS) proportional to changes in LV (LVEF) and RV function (TAPSE);Larger PALS changes and larger LA baseline volumes in the LARR+ group compared with LARR- group;Changes at multivariable analysis in the LA dimension (LAVi) proportional to changes in LVEDV and PALS.

In nearly one-third of patients treated with S/V, a positive LARR (reduction in LA volumes >15%) was also found after S/V treatment. That is in line with previous results by Castrichini et al. [26] and Mi-Gil Moon [30], which observed a further reverse remodeling induced by S/V on top of optimal medical treatment, not only on LV, but also on LA function.

Our data are in line with recent results by Sun et al. [31], which showed more significant benefits in terms of LV and LA reverse remodeling (defined as a reduction >15% in the LAVi) in the ARNI group vs. ACE/ARBs.

In patients with HFrEF, LA mechanical and neurohormonal anomalies are important factors involved in clinical deterioration, resulting in LA volume overload and decompensation [10]. A relationship between LA volume and function, expressed as deformation, has been found; larger LA volumes were correlated with lower PALS, defined as the main marker of LA deformation [32]. Impaired PALS values were associated with greater LV, worse values of either LVEF, LV global longitudinal strain (LV GLS) or RV systolic function, and more severe diastolic dysfunction [33]. 

S/V treatment might improve LV systolic and diastolic function [5] and reverse remodeling [6] in subjects with HFrEF. Therapy with S/V was also associated with an improved RV function in non-randomized studies [34].

In this study we did not only confirm and describe with more details the effect of S/V on LA function, but also identified possible predictors of LA remodeling after S/V therapy. In particular, at univariable analysis, principal predictors of LA remodeling in terms of changes in LAVi were systolic blood pressure values and LVEDD, while changes in LAVi were proportional to LVEDV and PALS.

According to such data, some functional hypotheses could be made. First, changes in LA volumes after S/V therapy are proportional to baseline LV dimensions. Second, changes in LA volumes are proportional to changes not only in LV volumes, but are also related to changes in LA function. Changes in LA volumes are mirrored by changes in LA function in terms of LA strain (PALS), which are linked to both LV (LVEF) and RV (TAPSE) function.

However, at multivariable analysis, no significant predictors of LA remodeling could be found, supporting the hypotheses that the effect of S/V therapy is generalized and not related with specific patient characteristics. The only borderline predictor of LA reverse remodeling seems to be LVEDD; the small number of patients enrolled in the study, however, does not allow any definitive conclusion. S/V was more effectively titrated in LA remodelers, probably due to significantly higher basal blood pressure values (Table 4).

Interestingly, changes in LA volumes are, even at multivariable analysis, paralleled by changes in LVEDV and PALS, suggesting a double functional path by which LA remodeling occurs. Probably, a strict connection between LV and LA function, volumes and strain might be supported by our preliminary data.

LA dimension and function play a role in the pathophysiology of HF and LA dysfunction may be the major driver/mediator of clinical decompensation in HF. There is an emerging role of LAEF and atrial strain parameters, such as PALS, LA conduit strain and PACS in predicting LA functional reverse remodeling, with a particular focus on PALS [35]. However, to the best of our knowledge, the role of PALS in predicting LARR in patients affected by HFrEF is not completely clear. In this study, we sought to characterize not only the predictive, but also the prognostic role of PALS and other advanced echocardiographic parameters in S/V-treated patients affected by HFrEF and on optimal medical therapy.

The improved LA function after S/V therapy found in our study is in line with data from Landolfo et al., who showed improved LA diameters after 12 months of S/V therapy [36]. Of note, our results are achievable already after just six months of therapy with S/V, confirming a quick benefit of the ARNI approach.

The E/e’ ratio and LV diastolic dysfunction were also improved, showing that a reduced LV loading after S/V therapy may be linked to an improved LA function. PALS and LA conduit turned out statistically significantly higher at follow up, with respect to baseline values, according to previous results [37]. We also found borderline improvements in PACS values. Our results are in accordance with those previously described by Castrichini et al. [26]; however, originally, we sought to describe possible predictors of LA remodeling and functional pathways involved in such remodeling. Some details on multi-chamber network interested by S/V therapy were thus identified; such preliminary data, however, needs to be confirmed in larger populations.

## 5. Conclusions

Six months of treatment with S/V in patients with HFrEF was associated with an improvement in LA functional reverse remodeling in a real-world scenario. LA reverse remodeling was not related to baseline echocardiographic variables, but was proportional to changes in LV volumes and LA strain. A strict connection between LA and LV reverse remodeling after S/V therapy and between LA anatomical and functional reverse remodeling seems to be outlined.

## 6. Limitations

This is a small, single center observational study, undersized for gender comparisons. Caution is required when these data are interpreted. The small number of patients enrolled in the study does not allow definitive conclusions: the study is underpowered for any comparison on variables, such as atrial fibrillation or hospitalizations between LARR+ vs. LARR−). With a type I error of 0.05 and a type II error (power) of 0.80, a ratio of 2:1 (LARR+ vs. LARR−), with the current population enrolled, a difference between rates of at least 35% could be detected as statistically significant.

## Figures and Tables

**Figure 2 jcm-12-01086-f002:**
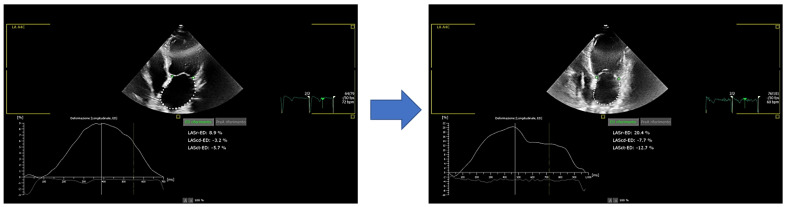
Left atrial strain improvement after 6-month therapy with Sacubitril/Valsartan. LASr-ED: left atrial strain reservoir phase at end-diastole, is the equivalent of PALS. LAScd-ED: left atrial strain conduit phase at end-diastole, is the equivalent of left atrial conduit strain. LASct: left atrial strain contraction phase at end-diastole, is the equivalent of PACS.

**Table 1 jcm-12-01086-t001:** Patients’ characteristics.

	Baseline	Follow Up	
	Mean ± SD or %	Mean ± SD or %	*p*-Value
Patients, #	47	47	
Male gender, %	85		
Age, years	66 ± 8		
BSA, m^2^	1.9 ± 0.2	1.9 ± 0.2	
Hypertension, %	85	85	
Diabetes, %	36	36	
Dyslipidemia, %	70	70	
Ischaemic etiology, %	43	43	
Atrial Fibrillation, %	51%	51%	
ICD/CRT-D, %/%	36/15	36/15	
NYHA class I, %	0	4	0.153
NYHA class II, %	64	79	0.107
NYHA class III, %	36	17	**0.037**
NYHA class IV, %	0	0	
SBP, mmHg	121 ± 17	116 ± 18	**0.01**
DBP, mmHg	71 ± 10	69 ± 10	0.163
HR, bpm	70 ± 13	69 ± 12	0.645
NT-proBNP, mg/dL	1086 ± 1116	399 ± 325	**<0.001**
Hb, g/dL	13 ± 1	13 ± 1	0.634
eGFR, mL/min/1.73 m^2^	71 ± 23	68 ± 24	0.103
Potassium, mEq/L	4.3 ± 0.2	4.5 ± 0.3	**0.006**
ACE-i, %	70	70	
ARBs, %	30	30	
Ivabradine, %	19	19	
MRAs, %	62	62	
Beta-blockers, %	98	98	

BSA: body surface area; LVEF: left ventricular ejection fraction; ICD: implantable cardioverter defibrillator; CRT-D: cardiac resynchronization therapy-defibrillator; NYHA: New York Heart Association; SBP: systolic blood pressure; DBP: diastolic blood pressure; HR: heart rate; NT-proBNP: N-terminal pro-brain natriuretic peptide; Hb: hemoglobin; eGFR: estimated glomerular filtration rate; Ace-i: angiotensin converting enzyme inhibitors; ARBs: angiotensin receptor blockers; MRAs: mineral corticoid antagonists. In bold *p* value < 0.05. Continuous variables are estimated by Student’s *t*-test.

**Table 2 jcm-12-01086-t002:** Standard echocardiographic parameters.

	Baseline	Follow Up	*p*-Value
	Mean ± SD	Mean ± SD	
LVEDDi, mm/m^2^	34.4 ± 5.0	33.7 ± 4.6	**0.002**
LVESDi, mm/m^2^	29.4 ± 5.1	28.1 ± 5.1	**<0.001**
LVEDVi, mL/m^2^	105 ± 34	99 ± 38	0.07
LVESVi, mL/m^2^	76 ± 28	65 ± 29	**<0.001**
LVEF, %	28 ± 6	36 ± 7	**<0.001**
LA area, cm^2^	24.7 ± 5.4	23.8 ± 5.6	**0.02**
LAVi, mL/m^2^	45 ± 15	42 ± 15	**0.008**
E/e’ ratio	16 ± 6	13 ± 6	**<0.001**
RAESA, cm^2^	20 ± 6	19 ± 6	0.219
PAsP	35 ± 9	31 ± 11	**0.001**
RVEDD, mm	38 ± 5	37 ± 5	**0.026**
TAPSE, mm	18 ± 3	20 ± 3	**<0.001**
RV TDI s’, cm/s	11 ± 2	12 ± 2	**<0.001**
RV FAC, cm^2^	34 ± 6	39 ± 6	**<0.001**

LVEDDi, left ventricular end-diastolic diameter indexed; LVESDi, left ventricular end-systolic diameter indexed; LVEDVi, left ventricular end-diastolic volume indexed; LVESVi, left ventricular end-systolic volume indexed; LVEF, left ventricular ejection fraction; LA area, left atrial area; LAVi, left atrial volume index; RAESA, right atrial end-systolic area; PAPs, systolic pulmonary artery systolic pressure; RVEDD, right ventricle end-diastolic diameter; TAPSE, tricuspid annular plane systolic excursion; RV TDI s’, right ventricle tissue Doppler imaging s’ wave; RVFAC, right ventricular fractional area change. In bold *p* value < 0.05. *p* values are estimated by Student’s *t*-test.

**Table 3 jcm-12-01086-t003:** Left atrial function parameters.

	Baseline	Follow Up	*p*-Value
	Mean ± SD	Mean ± SD	
LAEF, %	37 ± 11	46 ± 13	**<0.001**
PALS, %	15 ± 7	19 ± 8	**<0.001**
LA Conduit, %	(−7.6) ± 4.1	(−9.7) ± 5.2	**0.007**
PACS, %	8.7 ± 6	10.6 ± 6.5	0.051

LAEF: left atrial emptying fraction; PALS: peak atrial longitudinal strain; LA: left atrial; PACS: peak atrial contraction strain. In bold *p* value < 0.05. *p* values are estimated by Student’s *t*-test.

**Table 4 jcm-12-01086-t004:** Clinical and echocardiographic parameters in LARR+ and LARR− groups.

	LARR+ (N = 17)	LARR− (N = 30)	*p*-Value
	Mean ± SD	Mean ± SD	
Male gender, %	82	87	0.697
Age, years	66 ± 10	67 ± 7	0.722
BSA, m^2^	1.9 ± 0.2	1.9 ± 0.2	0.92
Hypertension, %	82	87	0.697
Diabetes, %	29	40	0.479
Dyslipidemia, %	65	73	0.545
Ischaemic etiology, %	35	47	0.46
Atrial fibrillation, %	35	67	0.108
ICD/CRT-D, %/%	29/12	40/17	
NYHA I, %	0	0	
NYHA II, %	59	67	0.583
NYHA III, %	41	33	0.583
NYHA IV, %	0	0	
SBP, mmHg	129 ± 21	117 ± 12	**0.012**
DBP, mmHg	74 ± 12	70 ± 9	0.226
HR, bpm	68 ± 11	71 ± 14	0.462
NT-proBNP, mg/dL ^a^	800(850)	800(1170)	0.631
LVEDVi, mL/m^2^	98 ± 27	109 ± 38	0.324
LVESVi, mL/m^2^	69 ± 21	80 ± 31	0.223
LVEF, %	30 ± 6	27 ± 5	0.148
LAVi, mL/m^2^	44 ± 17	46 ± 15	0.734
LAEF, %	40 ± 9	35 ± 11	0.143
PALS, %	14 ± 8	15 ± 7	0.872
E/e’ ratio	15 ± 7	16 ± 6	0.882
RA ESA, cm^2^	19 ± 3	20 ± 7	0.511
PAPs, mmHg	33 ± 9	36 ± 9	0.381
RVEDD, mm	39 ± 5	37 ± 5	0.362
TAPSE, mm	19 ± 3	18 ± 3	0.125
RV TDI s’, cm/s	11 ± 2	10 ± 2	0.078
RV FAC, cm^2^	37 ± 5	33 ± 6	**0.011**

LARR: left atrial reverse remodeling; SBP: systolic blood pressure; DBP: diastolic blood pressure; HR: heart rate; NT-proBNP: N-terminal pro-brain natriuretic peptide; LVEDVi: left ventricular end-diastolic volume indexed; LVESVi: left ventricular end-systolic volume indexed; LVEF: left ventricular ejection fraction; LAVi: left atrial end-systolic volume indexed; LAEF: left atrial emptying fraction; PALS: peak atrial longitudinal strain; RAESA: right atrial end-systolic area; PAPs: systolic pulmonary artery systolic pressure; RVEDD: right ventricle end-diastolic diameter; TAPSE: tricuspid annular plane systolic excursion; RV TDI s’: right ventricle tissue Doppler imaging s’ wave; RVFAC: right ventricular fractional area change. In bold *p* value < 0.05. *p* values are estimated by Student’s *t*-test. ^a^ median (interquartile range).

## Data Availability

The data that support the findings of this study are available from the corresponding author upon reasonable request.

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
