# Peer review of "Left Atrial Functional Remodeling in Patients with Chronic Heart Failure Treated with Sacubitril/Valsartan"

_jcm, 2023, doi:10.3390/jcm12031086_

Round 1

Reviewer 1 Report (Previous Reviewer 2)

Dear Sir/Madam,

I had the opportunity to act as a reviewer on the recent submission by Correale et al. to the Journal of Clinical Medicine.

The authors present an interesting paper studying left atrial echocardiographic parameters suggestive of atrial reverse remodeling in patients with heart failure with reduced ejection fraction (HFrEF) under therapy with valsartan/sacubitril. The key finding is that after six months of treatment with the substance the left atrial echocardiographic parameters improved.

The manuscript is very well structured and written. However, some issues need to be addressed: 

1.     Please provide baseline characteristics in patients with LARR+ versus patients with LARR- (table 4 with the same rows as table 1) in order to highlight potential differences between the groups.

2.     The multivariable analysis presented is obviously not robust enough for the small sample size (lines 304-309).

Best regards,

Author Response

Dear Sir/Madam,

I had the opportunity to act as a reviewer on the recent submission by Correale et al. to the Journal of Clinical Medicine.

The authors present an interesting paper studying left atrial echocardiographic parameters suggestive of atrial reverse remodeling in patients with heart failure with reduced ejection fraction (HFrEF) under therapy with valsartan/sacubitril. The key finding is that after six months of treatment with the substance the left atrial echocardiographic parameters improved.

The manuscript is very well structured and written. However, some issues need to be addressed: 

  1. Please provide baseline characteristics in patients with LARR+ versus patients with LARR- (table 4 with the same rows as table 1) in order to highlight potential differences between the groups.

Author’s response: Data were provided as required.

  1. The multivariable analysis presented is obviously not robust enough for the small sample size (lines 304-309).

               Author’s response: the limitation was acknowledged in the apposite section.

Reviewer 2 Report (New Reviewer)

Authors pay attention to very crucial problem of LA function, nowadays especially focused in present echocardiographic standards and trials. It is especially important in the terms of influence of LA function on clinical status of HFrEF patients. Focus on positive effects of pharmacological treatment on LA function and possible LA reverse remodeling play important role, especially in aspect of supraventricular arrythmias,( first of all atrial fibrillation incidence), that can deeply affect clinical status of HFrEF patients. LA function importance seems still to be underestimated, so the authors affect very crucial and practical problem, however ARNI treatment is now obvious base of HFrEF therapy. The analysis of echocardiographic parameters of LA, RV  and LV function in prediction of LA reverse remodeling is very useful and can be used in clinical practice.

The limitation , mentioned by authors, is small number of patients. The real world data analysis would be more valuable,  when the full standard treatment with SGLT inhibitors were applied, and the clinical parameters of LA reverse remodeling ( for example atrial fibrilation frequency) were evaluated.

Author Response

Review Report Form (Reviewer 2)

Authors pay attention to very crucial problem of LA function, nowadays especially focused in present echocardiographic standards and trials. It is especially important in the terms of influence of LA function on clinical status of HFrEF patients. Focus on positive effects of pharmacological treatment on LA function and possible LA reverse remodeling play important role, especially in aspect of supraventricular arrythmias, (first of all atrial fibrillation incidence), that can deeply affect clinical status of HFrEF patients. LA function importance seems still to be underestimated, so the authors affect very crucial and practical problem, however ARNI treatment is now obvious base of HFrEF therapy. The analysis of echocardiographic parameters of LA, RV and LV function in prediction of LA reverse remodeling is very useful and can be used in clinical practice.

The limitation, mentioned by authors, is small number of patients. The real world data analysis would be more valuable, when the full standard treatment with SGLT inhibitors were applied, and the clinical parameters of LA reverse remodeling (for example atrial fibrillation frequency) were evaluated.

Author’s response: We would thank our dear reviewer for his/her appreciation.

Reviewer 3 Report (New Reviewer)

Thank you for the opportunity to review this very interesting and potentially very practical manuscript. The study presented data indicating that the use of Sacubitril/Valsartan may affect the reverse functional remodeling of the left atrium. In this study, predictors of reverse functional remodeling of the left atrium were sought in the population of patients with HFrEF using S/V, but among the analyzed variables no parameters were identified that could indicate a good prognosis in terms of improved left atrial function. The advantage of the presented study is the use of the latest echocardiographic techniques to assess the function of the left atrium in the form of strain measurements in various phases of the left atrium. The disadvantage of the study is the relatively small study group (n=47), mainly men (85%). Unfortunately, the presented aim of the study does not correspond to the presented results.

Below are my comments:

1.       1. Lines 22 and 25 repeating the explanation of the abbreviation.

2.       Instead of LAESVi, the abbreviation LAVI - left atrial volume index is more often used in the literature and it is equivalent to LAESVi. I would recommend using the abbreviation LAVI.

3.       In the literature we use the abbreviation HFrEF, not CHFrEF.

4.       4. The same abbreviations are often repeated in the manuscript, e.g. LAEF, PALS, PACS, etc. Once an abbreviation is explained, there is no need to expand it again.

5.       Early diastolic velocity of mitral  annulus is e’, not E’.   

6.       Table S1 - tables are signed above and not below.

7.       Figure 1 - please provide units. It’s not clear.

8.       Line 193 - please specify what changes you mean. My guess is between the 6 month follow-up.

9.       Line 213 – TAPSE is not statistically significant.

10.   10. Table 4 and Table S2 are the same tables and the results in them are different.

11.   Line 216 „Evaluating S/V doses in the two subgroups  (LARR+  vs LARR-), we noted that the maximum dose of 97/103 mg at 6-months follow-up was higher in LARR+ than in LARR- patients (Figure S2). Please do a statistical analysis.

12.   Line 219 “…but the LARR+ group showed an improvement that was twice as high (55 ± 66 vs. 25 ± 26, p = 0.039). What is the result? What parameter? What units?

13.   Line 220 „PALS changes were higher according to LA dilation (mild, moderate or severe 14% vs. 32% vs 43% respectively)” Please do a statistical analysis.

14.  Line 266- it’s not the first. (1. JACC Cardiovasc Imaging. 2022 Sep;15(9):1525-1541; 2. J Clin Med. 2020 Mar 25;9(4):906.

15.    doi: 10.3390/jcm9040906.

16.   Line 268 – LAESVi is volume not dimension.

17.   Line 273 – Larger than what? You have not provided any statistical analysis.

18.   Line 274 – This “sentence” has no verb. What is this conclusion? Dimension is not volume!.

19.   Line 293-297 – you have not provided any data and analysis to support this paragraph. Further in line 304 you write that you have not shown anything.

Author Response

Review Report Form (Reviewer 3)

Thank you for the opportunity to review this very interesting and potentially very practical manuscript. The study presented data indicating that the use of Sacubitril/Valsartan may affect the reverse functional remodeling of the left atrium. In this study, predictors of reverse functional remodeling of the left atrium were sought in the population of patients with HFrEF using S/V, but among the analyzed variables no parameters were identified that could indicate a good prognosis in terms of improved left atrial function. The advantage of the presented study is the use of the latest echocardiographic techniques to assess the function of the left atrium in the form of strain measurements in various phases of the left atrium. The disadvantage of the study is the relatively small study group (n=47), mainly men (85%). Unfortunately, the presented aim of the study does not correspond to the presented results.

Below are my comments:

  1. Lines 22 and 25 repeating the explanation of the abbreviation.

Author’s response: Correction done.

  1. Instead of LAESVi, the abbreviation LAVI - left atrial volume index is more often used in the literature and it is equivalent to LAESVi. I would recommend using the abbreviation LAVI.

Author’s response: The whole text has been corrected.

  1. In the literature we use the abbreviation HFrEF, not CHFrEF.

Author’s response: Correction done.

  1. The same abbreviations are often repeated in the manuscript, e.g. LAEF, PALS, PACS, etc. Once an abbreviation is explained, there is no need to expand it again.

Author’s response: Correction done.

  1. Early diastolic velocity of mitral annulus is e’, not E’.

Author’s response: Correction done.

  1. Table S1 - tables are signed above and not below.

Author’s response: Correction done

  1. Figure 1 - please provide units. It’s not clear.

Author’s response: units were provided as required.

  1. Line 193 - please specify what changes you mean. My guess is between the 6 month follow-up.

Author’s response: Correction done.

  1. Line 213 – TAPSE is not statistically significant.

Author’s response: we agree with the reviewer's comment, we wrote that, among RV function parameters, TAPSE and RV TDI s’ showed only a positive trend, whilst RV FAC was statistically significant.

  1. Table 4 and Table S2 are the same tables and the results in them are different.

Author’s response: We deleted Table S2.

  1. Line 216 - Evaluating S/V doses in the two subgroups (LARR+ vs LARR-), we noted that the maximum dose of 97/103 mg at 6-months follow-up was higher in LARR+ than in LARR- patients (Figure S2). Please do a statistical analysis.

Author’s response: A p value was provided as required.

  1. Line 219 - but the LARR+ group showed an improvement that was twice as high (55 ± 66 vs. 25 ± 26, p = 0.039). What is the result? What parameter? What units?

Author’s response: The LARR+ group showed an improvement of the PALS that was twice as high (PALS changes: 55 ± 66 vs. 25 ± 26, p = 0.039).

  1. Line 220 - PALS changes were higher according to LA dilation (mild, moderate or severe 14% vs. 32% vs 43% respectively)” Please do a statistical analysis.

Author’s response: the line was deleted.

  1. Line 266- it’s not the first. (1. JACC Cardiovasc Imaging. 2022 Sep;15(9):1525-1541; 2. J Clin Med. 2020 Mar 25;9(4):906.

Author’s response: Both of these papers were cited as required.

  1. doi: 10.3390/jcm9040906.

Author’s response: we added the DOI in the reference number 26

  1. Line 268 – LAESVi is volume not dimension.

Author’s response: Correction done.

  1. Line 273 – Larger than what? You have not provided any statistical analysis.

Author’s response: Statistical analysis was provided.

  1. Line 274 – This “sentence” has no verb. What is this conclusion? Dimension is not volume!.

Author’s response: The section was rephrased as required.

  1. Line 293-297 – you have not provided any data and analysis to support this paragraph. Further in line 304 you write that you have not shown anything.

Author’s response: we found significant baseline predictors of LA remodeling but at multivariable analysis changes in LA volume were not proportional to any baseline parameter.

Round 2

Reviewer 3 Report (New Reviewer)

Congratulations on the article

This manuscript is a resubmission of an earlier submission. The following is a list of the peer review reports and author responses from that submission.

Round 1

Reviewer 1 Report

In this study, Dr. Correale and colleagues explored the usefulness of ARNI (Sacubitril/Valsartan) to reverse atrial remodeling in chronic HFrEF patients. Overall, although the topic is relevant to explore, there are some issues with this manuscript, requiring clarifications and revisions:

  • I strongly suggest to improve the readability and clarity of this manuscript. This manuscript is not easy to read and lack of coherence between sentences. Some typographical errors were also found throughout the manuscript.
  • I also suggest to avoid using bullets and numbering in the text. Instead, the authors could improve the narration.
  • "Previous studies showed left ventricular (LV) and left atrial (LA) improvement and reverse remodeling after therapy with Sacubitril/Valsartan (S/V)." in which disease or pathological condition? Please specify
  • "...investigate predictors and correlates of LA structural and functional remodeling in patients with chronic heart failure and reduced LVEF after therapy with S/V" What does it mean by "predictors and correlates"? What did the authors tried to investigate? Please rephrase. This is also a bit different than that written in line 84 "The aim of this study was therefore to analyze the predictive and the prognostic role of PALS and other parameters in predicting LARR patients with CHFrEF treated with S/V." Please be consistent.
  • "A significant reduction of LAEF (46 ± 13% vs. 37 ± 11%, p < 0.001) and a non-significant reduction of LAESVi (42 ± 15 vs. 45 ± 15 mL/m2, p = 0.08) were found at follow up" I don't understand this statement. There are only two time points in this study: baseline and 6 months after S/V. So, this indicates that the LAEF was reduced after S/V? Please clarify.
  • Also, in Table 3, the LAEF was increased in follow up. So, which one is true?
  • Also, the authors said that the LAESVi was non-significantly reduced at follow up, which was not correctly reported in Table 2. In table 2, it is clear that the p-value was significant (0.008). Which one is true?
  • These two examples could indicate that this manuscript contain many data inaccuracy. I strongly suggest to recheck everything again, including all the p-values and please re-do the statistical analyses to check since some results are weird (I will show them below). 
  • "LA strain parameters, PALS (15 ± 7 vs. 19 ± 8 %, p < 0.001) and LA conduit (-7.6 ± 4.1 vs. -9.7 ± 5.2%, p = 0.007), significantly improved." When? at follow-up?
  • Line 43: "increased rigidity of muscular tissue" what does this mean? Maybe it should be "stiffness"? Also, which muscular tissue?
  • Regarding the information in lines 52-73 about the reported effects of S/V, I suggest the authors to read and add some data from this publication (PMID: 34445698)
  • Line 68: what is "NPS"?
  • "Patients were therefore divided in two groups, those with a LARR (relative reduction in LA end-systolic volume indexed (LAESVi) > 15% [25], LARR+) and those without (LARR-)." This was not stated in the abstract at all. Why? Please add this information about LARR in the abstract. If needed, the abstract should be restructured to better reflect the study design and aims. 
  • Please add a flowchart of study design with the details on number of samples at each steps, interventions and inclusion/exclusion criteria.
  • Consider adding some echo (also speckle tracking) images comparing LARR + and -, also between baseline and follow-up.
  • In Section 2.3, what does it mean by "as required"? Please be specific, when was it required?
  • "34% patients were titrated with the maximum dose of S/V 97/103 mg b.i.d., and 30% patients were in therapy with 49/51 mg b.i.d. after six months of treatment." Why did some of the patients receive higher dose than the others? Please explain. Also, 34% + 30% = 64%, so where are the rest? Which dose did the rest of patients receive and why? 
  • Also, please explain, when was the titration started and how was it done?
  • Please show all the scatter plots for Section 3.3 about correlations.
  • I think the authors should move Figure S2 to the main manuscript.
  • "No significant difference regarding basal echocardiographic parameters was found between the two groups" But the RV FAC was significantly different. Please clarify. 
  • I found that the data in Table 1 is a bit odd. S/V is known to lower BP. In fact, Sacubitril is a potent antihypertensive drug. Why then the percentage of HT patients remained unchanged at follow up? This is bizzare. Please comment on this.
  • Please also add the number of NYHA class I and IV at follow-up.
  • Why was the potassium level significantly increased at follow-up? Could the authors explain the mechanism behind this observation? 
  • In Table 2, there are also weird data. For example, the LVEDDi were identical (34±5) but there was a statistical significance. How could this happen? Please re-perform all the statistical analyses. I don't think I can trust them now. 
  • This also happens with LVESDi and LA area. There was only a slight difference, which will be compensated by the SD but the p-value was highly significant (<0.001 and 0.02). How? I doubt it. Please consult with a statistician if needed.
  • Please recheck the caption of Table 2, "LAEF, left atrial emptying fraction" was specified but there was no LAEF in Table 2. 
  • "In this study we did not only confirmed and described with more"
  • "particular classes of patients." is this NYHA class? If so, how did the authors know if they only included NYHA class II and III?

In general, the authors need to improve the quality of the writing and the robustness (and accuracy) of their analyses and data. Only by then, we can discuss the content of this manuscript in more details.

Reviewer 2 Report

Dear Sir/Madam,

I had the opportunity to act as a reviewer on the recent submission by Correale et al. to the Journal of Clinical Medicine.

The authors present interesting original research regarding left atrial function remodeling in patients with heart failure under therapy with valsartan/sacubitril. They have included a total of 47 patients and found that after six months of treatment with valsartan/sacubitril the left atrial functional remodeling improved.

The manuscript is well written and the results are interesting and of high clinical interest.

However, two major issues need to be addressed:

  1. I recommend presenting a clear power calculation in the Methods part. This seems to be missing and the sample size of 47 patients is rather low. There is however a short explanation in the Limitations part (lines 310-312), but I recommend improving and moving it into the Methods.
  1. The last sentence on page 4, lines 189-193 refers to the multivariable regression analysis, which seems to include the following parameters: age, gender, baseline NT-proBNP levels, drug doses after titration, systolic blood pressure, baseline LVEDD and LVESD values, changes in RV TDI s’, LVEDV, LVESV and PALS values. There are 11 predictors for a sample size of 47. This is clearly insufficient: I recommend either raising the number of patients (i.e., 110-120) or reducing the number of predictors to 4, maximum 5.

Furthermore, in Table S1 the reader does not find the expected predictors, only three of them. In the Discussion section, lines 258-260, the reader learns that the principal predictors of LA remodeling in terms of changes in LAESVi were systolic blood pressure values and LVEDD, while changes in LAESVi were proportional to LVEDV and PALS.

This is very confusing, please explain.

Minor issues:

  1. Please mention that the therapy with SGL2i was not yet recommended in the HF guidelines at the time of recruitment.

Best regards,